
# Trapping effects in quantum atomic arrays

**Pengfei Zhang**

Institute for Quantum Information and Matter and Walter Burke Institute for Theoretical Physics, California Institute of Technology, Pasadena, CA 91125, USA

⋆ PengfeiZhang.physics@gmail.com

## Abstract

Quantum emitters, particularly atomic arrays with subwavelength lattice constant, have been proposed to be an ideal platform for studying the interplay between photons and electric dipoles. In this work, motivated by the recent experiment [1], we develop a microscopic quantum treatment using annihilation and creation operator of atoms in deep optical lattices. Using a diagrammatic approach on the Keldysh contour, we derive the cooperative scattering of the light and obtain the general formula for the $S$ matrix. We apply our method to study the trapping effect, which is beyond previous treatment with spin operators. If the optical lattices are formed by light fields with magical wavelength, the result matches previous results using spin operators. When there is a mismatch between the trapping potentials for atoms in the ground state and the excited state, atomic mirrors become imperfect, with multiple resonances in the optical response. We further study the effect of recoil for large but finite trapping frequency. Our results are consistent with existing experiments.

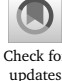

## 1   Introduction

The ability to coherently storing photons and controlling their interaction with quantum matters is of vital importance for quantum science. Although single atoms and photons usually interact less efficiently, ensembles of atoms can show a cooperative response of photons. As an example, superradiance can be realized when the radiations between atoms interfere constructively [2–8]. Recently, atomic arrays with subwavelength lattice structures are found to be an ideal platform where electric dipole-dipole interactions between atoms are mediated by photons [9–18]. The analysis shows the atomic arrays exhibit subradiance and are nearly perfect mirrors for a wide range of incident angles [16], as observed in recent experiments [1]. Later, there are many theoretical studies on the fruitful physics in atomic arrays [19–26, 26–41]. For example, there are proposals for realizing non-trivial topology in atomic arrays [19–21], controlling atom-photon interaction using atomic arrays [22–26], and efforts in understanding their subradiant behaviors and ability of photon storage [26–32].

In most of these works, atoms are treated as point-like with no motional degree of freedom. The evolution of the system is described by using non-Hermitian Hamiltonian or Lindblad master equation [16, 17], with spin degree of freedom $\sigma_{im}^- = |\mathbf{r}_m, g\rangle\langle\mathbf{r}_m, e_i|$. Here $|\mathbf{r}_m, g\rangle$ is the $s$-wave ground state for the atom at position $\mathbf{r}_m$. $|\mathbf{r}_m, e_i\rangle$ is the $p$-wave excited labeled by the dipole moment $\mathbf{d} = d\,\mathbf{e}_i$ of the corresponding transition $g \to e_i$. However, in real experiments, the system consists of atoms moving in optical lattices. For deep optical lattices, although atoms are trapped near the potential minimum, the wave function for the motional degree of freedom may still play a role. Moreover, the consequence of fractional filling has been studied in the experiment. It is difficult to analyze the absence of an atom in the spin-operator language, and consequently, theoretical predictions for the fractional filling case are still absent.

In this work, we overcome this difficulty by using a microscopic model for the coupled system consisting of atoms in deep optical lattices and photons. After making plausible assumptions, we derive the cooperative response of the system using a diagrammatic approach on the Keldysh contour. By summing up bubble diagrams with dressed Green's function, we obtain neat results for the transmission coefficient and the reflection coefficient, with the contribution from the motional wave function. Our result matches the previous analysis for unit filling when the potential of the excited state atoms are the same as that of the ground state. We study the effect of the discrepancy of optical lattices for the ground state and excited state atoms, where the transition of the internal state can be accompanied by transitions in the motional degree of freedom. In particular, we find that multiple resonances can exist in the response function. The cooperative linewidth is linear in $n$, consistent with the experimental observation and previous theoretical predictions [38]. We further study the effect of recoil for large but finite trapping frequency.

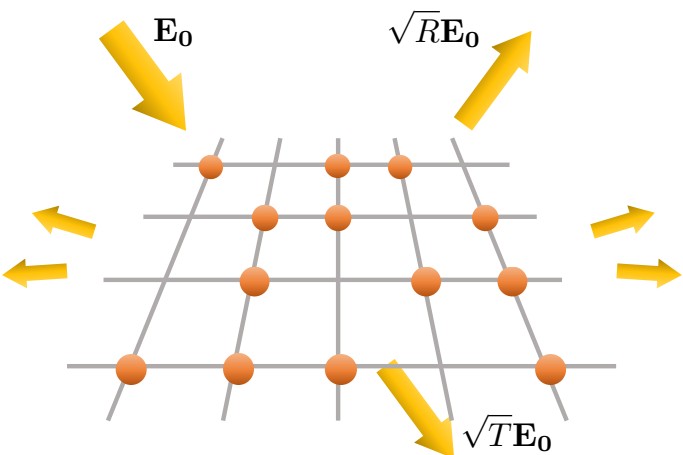

Figure 1: Schematics of the model considered in this work: the atomic array in the optical lattices at fractional filling.

## 2 Diagrammatic Approach to Quantum Atomic Arrays

### 2.1 Model

We consider coupled systems with atoms and photons. Th Hamiltonian reads

$$H = H_{\text{EM}} + H_{\text{A}} + H_{\text{int}}. \tag{1}$$

Here the first term is the Hamiltonian of the electromagnetic field

$$H_{\text{EM}} = \int d\mathbf{r} \left( \frac{\epsilon_0}{2} \mathbf{E}(\mathbf{r})^2 + \frac{\mu_0}{2} \mathbf{H}(\mathbf{r})^2 \right). \tag{2}$$

The second term describes the motion of atoms in optical lattices

$$H_{\text{A}} = \int d\mathbf{r} \sum_i \psi_{e_i}^\dagger(\mathbf{r}) \left( \omega_0 - \frac{\nabla^2}{2} + V_{e_i}(\mathbf{r}) \right) \psi_{e_i}(\mathbf{r}) + \int d\mathbf{r}\, \psi_g^\dagger(\mathbf{r}) \left( -\frac{\nabla^2}{2} + V_g(\mathbf{r}) \right) \psi_g(\mathbf{r}). \tag{3}$$

$V_{g/e_i}(\mathbf{r})$ describes the optical lattice potential for ground/excited-state atoms. We have set $\hbar = 1$ and $m = 1$. We assume each site is occupied by at most one atom, which corresponds to choosing fermionic commutation relation $\{\psi_a^\dagger(\mathbf{r}), \psi_b(\mathbf{r}')\} = \delta_{ab}\delta(\mathbf{r} - \mathbf{r}')$. The last term describes the interaction between atoms

$$H_{\text{int}} = -\int d\mathbf{r} \sum_i \left( P_i^+(\mathbf{r}) + P_i^-(\mathbf{r}) \right) \mathbf{e}_i \cdot \mathbf{E}(\mathbf{r}). \tag{4}$$

Here $\mathbf{e}_i$ is the unit polarization vector along the $i$ direction and

$$P_i^+(\mathbf{r}) = d\, \psi_{e_i}^\dagger(\mathbf{r})\psi_g(\mathbf{r}) = P_i^-(\mathbf{r})^\dagger. \tag{5}$$

The full Hamiltonian (1) describes general model with interaction between atoms and light to the order of electric dipole transition. For atomic arrays, the ground state particle is always tightly trapped near the local minimum of optical lattices, with a spread of wave function $\sigma \ll a_0$, where $a_0$ is the lattice constant [1]. Assuming the excited state is also deeply trapped, we expand

$$\psi_g(\mathbf{r}) \approx \sum_{n,a} \varphi_a(\mathbf{r} - \mathbf{r}_n)\psi_g^a(\mathbf{r}_n), \qquad \psi_{e_i}(\mathbf{r}) \approx \sum_{n,a} \varphi'_{i,a}(\mathbf{r} - \mathbf{r}_n)\psi_{e_i}^a(\mathbf{r}_n). \tag{6}$$

Here $\varphi_a(\mathbf{r})/\varphi'_{i,a}(\mathbf{r})$ is the motional wave function for ground/excited-state atoms near the local minimum $\mathbf{r}_n = \mathbf{0}$ with the energy $\varepsilon_a/\varepsilon'_{i,a}$. We neglect the tunneling between different sites, which is suppressed expoenentially and as a result the local motional wave function conincident with the Wannier function. We have $\mathbf{r}_n = a_0(n_1\mathbf{e}_1 + n_2\mathbf{e}_2)$, where we use a single index $n$ to represent $(n_1, n_2)$ for conciseness. The commutation relation for $\psi^a_\eta(\mathbf{r}_n)$ now becomes $\{\psi^{a,\dagger}_\eta(\mathbf{r}_m), \psi^b_\xi(\mathbf{r}_n)\} = \delta_{\eta\xi}\delta_{ab}\delta_{mn}$. Using (6), the Hamiltonian $H_A$ and $H_{\text{int}}$ can be simplified. We have

$$H_A = \sum_{n,a}\left[\sum_i \varepsilon'_{i,a}\psi^{a,\dagger}_{e_i}\psi^a_{e_i}(\mathbf{r}_n) + \varepsilon_a\psi^{a,\dagger}_g\psi^a_g(\mathbf{r}_n)\right], \tag{7}$$

and (4) becomes

$$H_{\text{int}} = -\sum_{n,i}\left(p^+_i(\mathbf{r_n})\,\mathbf{e}_i\cdot\mathbf{E}(\mathbf{r_n}) + \text{H.C.}\right), \tag{8}$$

with

$$p^+_i(\mathbf{r_n}) = d\sum_{ab}\int d\mathbf{r}\,\varphi'_{i,a}(\mathbf{r})^*\varphi_b(\mathbf{r})\,\psi^{a,\dagger}_{e_i}(\mathbf{r}_n)\psi^b_g(\mathbf{r}_n). \tag{9}$$

Equation (2), (7) and (8) describe the dynamics of the atomic array. Initially, we prepare all atoms in the $s$-wave internal ground state $|g\rangle$ with motional ground state $\varphi_0(\mathbf{r})$. The number of atoms in the excited states are suppressed due to the violation of energy conservation. We further add an external probe light, at fixed frequency $\omega$, which is near-resonant with $\delta \equiv \omega - \omega_0 \ll \omega, \omega_0$ [1]. The electric field reads $\mathbf{E}_0(\mathbf{r}) = \mathbf{E}_0 e^{i\mathbf{k}\cdot\mathbf{r}}$ with $c|\mathbf{k}| = \omega$. We take $c = 1$ from now on for conciseness. This probe corresponds to the incident light in the scattering experiment. Its coupling to atoms reads

$$\delta H = -\sum_{n,i}\left(p^+_i(\mathbf{r_n})e^{-i\omega t}\mathbf{e}_i\cdot\mathbf{E}_0(\mathbf{r}_n) + \text{H.C.}\right). \tag{10}$$

We assume the field strength $\mathbf{E}_0$ is weak and the response can be analyzed using the linear response theory. The total electric field including the incident light and the scattered light then reads

$$\mathbf{E}_{\text{tot}}(\omega, \mathbf{r}) = \mathbf{E}_0(\mathbf{r}) + \langle\mathbf{E}(\omega, \mathbf{r})\rangle. \tag{11}$$

Far from the atomic array, when only a single diffraction order exists, we expect

$$\mathbf{E}_{\text{tot}}(\omega, \mathbf{r}) = \left(\mathbf{1}e^{ik_z z} + \mathbf{S}(\omega, \mathbf{k}_\parallel)e^{ik_z|z|}\right)\cdot\mathbf{E}_0 e^{i\mathbf{k}_\parallel\cdot\mathbf{r}_\parallel}, \tag{12}$$

and $\mathbf{S}(\omega, \mathbf{k}_\parallel)$ is the corresponding $\mathbf{S}$ matrix.

## 2.2 Diagrammatic Expansion

The contribution to the scattered light $\langle\mathbf{E}(\omega, \mathbf{r})\rangle$ can be efficiently organized using the path-integral formulism. In particular, we work on the Keldysh contour [42], which contains a forwardly evolving branch and a backwardly evolving branch, corresponding to $e^{-iHt}$ and $e^{iHt}$ in the Heisenberg evolution. It is one of standard techniques for analyzing quantum many-body dynamics and systems with disorders.

---

[1]As a result, we will not distinguish $\omega$ and $\omega_0$ unless they combine into $\delta$.

The expectation of fluctuation field becomes non-zero due to the coupling to atoms. Diagrammatically, we have

$$\langle \mathbf{E}(\omega, \mathbf{r}) \rangle = \overset{\mathbf{E}(\omega,\mathbf{r}) \qquad \mathbf{p}^-(\omega,\mathbf{r}_n)}{\sim\!\!\sim\!\!\sim\!\!\sim\!\!\bullet}$$
$$= -\sum_n \mathbf{G}_\mathrm{R}^\mathbf{E}(\omega, \mathbf{r} - \mathbf{r}_n) \cdot \langle \mathbf{p}^-(\omega, \mathbf{r}_n) \rangle. \tag{13}$$

Here we use the wavy line to represent the propagation of photons. $\mathbf{G}_\mathrm{R}^\mathbf{E}$ is the retarded Green's function matrix of $\mathbf{E}$ in free space defined as

$$\mathbf{G}_\mathrm{R}^\mathbf{E}(t, \mathbf{r}) \equiv -i\theta(t) \langle [E(t, \mathbf{r}), E(0, \mathbf{0})] \rangle_{d=0}. \tag{14}$$

In frequency and momentum space, we have

$$\tilde{\mathbf{G}}_\mathrm{R}^\mathbf{E}(\omega, \mathbf{k}) = (\epsilon_0 \mathbf{1} + \epsilon_0 \mathbf{k} \times \mathbf{k} \times /\omega^2)^{-1} = -\frac{\omega^2}{\epsilon_0} \tilde{\mathbf{G}}(\omega, \mathbf{k}). \tag{15}$$

Here $\tilde{\mathbf{G}}(\omega, \mathbf{k})$ is the standard dyadic Green's function [16, 43]. Note that we have added an additional tilde for the Green's function of photons in momentum space to avoid possible confusion. The local dipole moment $\mathbf{p}^-$ is related to the incident light $\mathbf{E}_0$ by the Kubo formula [44]

$$\langle \mathbf{p}^-(\omega, \mathbf{r_n}) \rangle = \overset{\mathbf{p}^-(\omega,\mathbf{r}_n) \qquad \mathbf{p}^+(-\omega,\mathbf{r}_m)}{\rule[0.5ex]{4cm}{1pt}}$$
$$= -\sum_m \mathbf{G}_\mathrm{R}^\mathbf{p}(\omega, \mathbf{r}_{nm}) \cdot \mathbf{E}_0(\omega, \mathbf{r}_m). \tag{16}$$

We use the double solid line for the retarded Green's function for dipole momentums $\mathbf{G}_\mathrm{R}^\mathbf{p}(\omega, \mathbf{r})$ and $\mathbf{r}_{nm} \equiv \mathbf{r}_n - \mathbf{r}_m$. This is consistent with the semi-classical analysis [16]. The remaining task is to derive approximate formula for $\mathbf{G}_\mathrm{R}^\mathbf{p}(\omega, \mathbf{r})$, which includes renormalization due to the coupling with photons.

In this work, we take diagrams with single excitation which conserves the total energy. We first consider the correction of the excited state Green's function $G_\mathrm{R}^{e_i}(\omega, \mathbf{r}, \mathbf{r}')$ by emission and absorption of photons. As we will see, since the wave function for ground-state atoms is localized, only $G_\mathrm{R}^{e_i}(\omega, \mathbf{r}, \mathbf{r}')$ with $\mathbf{r} \approx \mathbf{r} \approx \mathbf{r}_n$ contributes to the light scattering. The bare Green's function near $\mathbf{r}_n = \mathbf{0}$ reads

$$G_\mathrm{R}^{0,e_i}(q_0, \mathbf{r}, \mathbf{r}') \approx \sum_a \frac{\varphi'_{i,a}(\mathbf{r})\varphi'_{i,a}(\mathbf{r}')^*}{q_0 - \omega_0 - \varepsilon'_{i,a} + i0^+}. \tag{17}$$

The Schwinger-Dyson equation reads $(G_\mathrm{R}^{e_i})^{-1} = (G_\mathrm{R}^{0,e_i})^{-1} - \Sigma_\mathrm{R}^{e_i}$, with the self-energy $\Sigma_\mathrm{R}^{e_i}$

$$\Sigma_\mathrm{R}^{e_i}(q_0, \mathbf{r}, \mathbf{r}') = \overset{\overgroup{\sim\!\!\sim\!\!\sim}}{\underset{\mathbf{E} \qquad e_i}{\longleftarrow\!\!\bullet\!\!\longleftarrow}}$$
$$\approx -\frac{\omega^2 d^2}{\epsilon_0} G_{ii}(\omega, \mathbf{0}) \sum_a (1 - n_a) \varphi_a(\mathbf{r}) \varphi_a^*(\mathbf{r}'). \tag{18}$$

Here $n_{a \geq 1} = 0$ and $n_0 = n$ is equal to the filling fraction. The appearance of $G_{ii}(\omega, \mathbf{0}) = \mathbf{e}_i \cdot \mathbf{G}(\omega, \mathbf{0}) \cdot \mathbf{e}_i$ owes to the approximation in (8) by using $\mathbf{E}(\mathbf{r}_n)$ instead of $\mathbf{E}(\mathbf{r})$. The real-part of $\mathbf{G}(\omega, \mathbf{0})$ contributes to the lamb shift, which can be absorbed in the definition of $\omega_0$. As a result, we only keep the imaginary part $\mathbf{G}(\omega, \mathbf{0}) = i\omega/6\pi \times \mathbf{1}$. We also assume $\delta, \varepsilon_a, \varepsilon'_{i,a} \ll \omega$, and the resonance frequency $\omega$ is much larger than the loop frequency, which

is an analogy of the Markovian approximation in the master equation [16]. Note that in the path-integral approach, the Green's function is defined by adding an addtional particle on top of the many-body system with filling $n$, as a result, the Pauli exclusion principle exists and contributes to the $(1 - n_a)$ factor. The natural linewidth of a transition with frequency $\omega$ is known to be $\gamma = \omega_0^3 d^2 / 3\pi\epsilon_0$. This leads to

$$\Sigma_{\mathrm{R}}^{e_i}(q_0, \mathbf{r}, \mathbf{r}') \approx -\frac{i\gamma}{2} \left[ \delta(\mathbf{r} - \mathbf{r}') - n\varphi_0(\mathbf{r})\varphi_0^*(\mathbf{r}') \right], \tag{19}$$

where we have used the completeness of local wave functions $\sum_a \varphi_a(\mathbf{r})\varphi_a^*(\mathbf{r}') = \delta(\mathbf{r} - \mathbf{r}')$.

Having obtained the dressed Green's function, we turn to the calculation of $\mathbf{G}_{\mathrm{R}}^{\mathrm{p}}(\omega, \mathbf{r})$. Motivated by the standard Random Phase Approximation (RPA) in interacting fermions [45], we consider the diagrams

$$\mathbf{G}_{\mathrm{R}}^{\mathrm{p}} = \underset{g}{\overset{e_i}{\langle \delta_{mn} \rangle}} + \underset{g}{\overset{e_i}{\langle \mathbf{r_n} \rangle}} \sim \underset{g}{\overset{e_j}{\langle \mathbf{r_m} \rangle}} \dots \tag{20}$$

Note that in our diagrammatic approach, $\mathbf{r}_n$ can be equal to $\mathbf{r}_m$, which is important as we will see later. The thick solid line represents the normalized Green's function $G_{\mathrm{R}}^{e_i}$. The first bubble diagram, which is an elementary building block, is given by

$$i[\mathbf{\Pi}_{\mathrm{R}}(\omega)]_{ij} = \frac{d^2}{2} \int \frac{dq_0}{2\pi} d\mathbf{r}' d\mathbf{r} \left[ G_{\mathrm{R}}^{e_i}(q_0 + \omega, \mathbf{r}, \mathbf{r}') G_{\mathrm{K}}^g(q_0, \mathbf{r}', \mathbf{r}) + G_{\mathrm{K}}^{e_i}(q_0 + \omega, \mathbf{r}, \mathbf{r}') G_{\mathrm{A}}^g(q_0, \mathbf{r}', \mathbf{r}) \right] \delta_{ij}. \tag{21}$$

Here we write $\mathbf{\Pi}_{\mathrm{R}}(\omega)$ as a diagonal matrix for later convenience. $G_{\mathrm{A}}^\eta$ is the advanced Green's function. $G_{\mathbf{K}}^\eta = G_{\mathrm{R}}^\eta \circ F_\eta - F_\eta \circ G_{\mathrm{A}}^\eta$ is the Keldysh Green's function, and $F_\eta = (1 - 2n_\eta)$ is the quantum distribution function [42]. Here we use $\circ$ to represent the inner product of functions by integration. This leads to

$$\mathbf{\Pi}_{\mathrm{R}}(\omega)_{ii} = d^2 n \int d\mathbf{r}' d\mathbf{r} \, G_{\mathrm{R}}^{e_i}(\omega + \varepsilon_0, \mathbf{r}, \mathbf{r}')\varphi_0(\mathbf{r})^*\varphi_0(\mathbf{r}') = d^2 n \, \varphi_0^* \circ G_{\mathrm{R}}^{e_i} \circ \varphi_0. \tag{22}$$

It can be further simplified by noticing that

$$\varphi_0^* \circ G_{\mathrm{R}}^{e_i} \circ \varphi_0 = \sum_a \frac{(\varphi_0 \circ \varphi_{i,a}')^* \, \varphi_{i,a}' \circ \varphi_0}{\delta + \varepsilon_0 - \varepsilon_{i,a}' + \frac{i\gamma}{2}} + \frac{i\gamma n}{2} \left( \sum_a \frac{(\varphi_0 \circ \varphi_{i,a}')^* \, \varphi_{i,a}' \circ \varphi_0}{\delta + \varepsilon_0 - \varepsilon_{i,a}' + \frac{i\gamma}{2}} \right)^2 + \dots \tag{23}$$

As a result, we have

$$\mathbf{\Pi}_{\mathrm{R}}(\omega)_{ii} = \frac{d^2 n}{\pi_i(\omega)^{-1} - i\frac{\gamma n}{2}}, \qquad \pi_i(\omega) \equiv \sum_a \frac{(\varphi_0 \circ \varphi_{i,a}')^* \, \varphi_{i,a}' \circ \varphi_0}{\delta + \varepsilon_0 - \varepsilon_{i,a}' + \frac{i\gamma}{2}}. \tag{24}$$

Then we can sum over the diagrams with multiple bubbles in (20). This gives

$$\begin{aligned} i\mathbf{G}_{\mathrm{R}}^{\mathrm{p}}(\omega, \mathbf{k}_\parallel) &= i\mathbf{\Pi}_{\mathrm{R}}(\omega) - i\mathbf{\Pi}_{\mathrm{R}}(\omega) i\tilde{\mathcal{G}}_{\mathrm{R}}^{\mathrm{E}}(\omega, \mathbf{k}_\parallel) i\mathbf{\Pi}_{\mathrm{R}}(\omega) + \dots \\ &= \frac{i}{\mathbf{\Pi}_{\mathrm{R}}(\omega)^{-1} - \tilde{\mathcal{G}}_{\mathrm{R}}^{\mathrm{E}}(\omega, \mathbf{k}_\parallel)}. \end{aligned} \tag{25}$$

Since the summation in (16) is descrete, the Fourier transform here is defined as

$$\tilde{\mathcal{G}}_{\mathrm{R}}^{\mathrm{E}}(\omega, \mathbf{k}_\parallel) = \sum_n G_{\mathrm{R}}^{\mathrm{E}}(\omega, \mathbf{k}_\parallel) e^{-i\mathbf{k}_\parallel \cdot \mathbf{r}_n}. \tag{26}$$

In particular, the denominator of (25) is a generalization of the corresponding result under non-Hermitian Hamiltonians, which is $\omega\mathbf{1}-\mathcal{H}_{\text{eff}}$. As we will see later, (21) takes such a form only for unit filling and $V_{e_i}(r) = V_g(r)$. This implies the breakdown of non-Hermitian Hamiltonian description for general setups.

Then, using the relation (15), we obtain the relation between $\langle\mathbf{p}^-\rangle$ and $\mathbf{E}_0$ in momentum space as

$$\begin{aligned}
\langle\mathbf{p}^-(\omega,\mathbf{k}_\parallel)\rangle &= \boldsymbol{\alpha}(\omega,\mathbf{k}_\parallel)\cdot\mathbf{E}_0(\mathbf{k}_\parallel),\\
\boldsymbol{\alpha}(\omega,\mathbf{k}_\parallel)^{-1} &= -\mathbf{\Pi}_\text{R}(\omega)^{-1}+\tilde{\mathcal{G}}_\text{R}^\mathbf{E}(\omega,\mathbf{k}_\parallel).
\end{aligned}\tag{27}$$

Finally, for a single diffraction order, using (13), $\boldsymbol{\alpha}$ is related to the $\mathbf{S}$ matrix as [16]

$$\mathbf{S}(\omega,\mathbf{k}_\parallel)=\frac{i\omega^2}{2a^2\epsilon_0 k_z}\mathcal{P}(\omega,\mathbf{k}_\parallel)\cdot\boldsymbol{\alpha}(\omega,\mathbf{k}_\parallel).\tag{28}$$

Here $\mathcal{P}_{ij}(\omega,\mathbf{k}_\parallel)=\delta_{ij}-\xi_{ij}\frac{k_i k_j}{\omega^2}$. $\xi_{ij}=-1$ if only one of $(i,j)$ is in $z$ direction, and at the same time $z < 0$. In other cases, $\xi_{ij}=1$.

Further simplification is possible for the normal incident case as in the experiment [1], where we have $k_x = k_y = 0$, $k_z = \omega$. Since, $\mathbf{E}_0$ lies in the $x$-$y$ plane, we have $\mathcal{P}=\mathbf{1}$. Moreover, due to the rotational symmetry, $\tilde{\mathcal{G}}_\text{R}^\mathbf{E}(\omega,\mathbf{0})$ is also a diagonal matrix. Following the convention [16], we define $\mathbf{\Delta}(\mathbf{k}_\parallel)=-\frac{3\pi\gamma}{\omega}\sum_{n\neq 0}\text{Re}\,\boldsymbol{G}(\omega,\mathbf{r}_n)e^{-i\mathbf{r_n}\cdot\mathbf{k}_\parallel}$ and $\mathbf{\Gamma}(\mathbf{k}_\parallel)=\frac{6\pi\gamma}{\omega}\sum_{n\neq 0}\text{Im}\,\boldsymbol{G}(\omega,\mathbf{r}_n)e^{-i\mathbf{r_n}\cdot\mathbf{k}_\parallel}$, which also become scalars $\Delta$ and $\Gamma$ in the $x$-$y$ plane for normal incident light. In particular, it is known that $\Gamma + \gamma = \gamma\frac{3\pi}{a^2\omega^2}$ [16]. As a result, the $\mathbf{S}$ matrix is diagonal and there is no mixing between contributions from different excited states $e_i$. For the incident light polarized in $i_0$ direction, the only relevant response function is $\Pi_\text{R}(\omega)\equiv\Pi_{\text{R},i_0 i_0}(\omega)$, which is related to the local response $\pi(\omega)\equiv\pi_{i_0}(\omega)$. We also drop the $i_0$ index in $\varepsilon'_a\equiv\varepsilon'_{i_0,a}$ and $\varphi'_a\equiv\varphi'_{i_0,a}$ for conciseness. From now on, we focus on this normal incident case unless mentioned otherwise.

Using these definitions, we have

$$\pi(\omega)=\sum_a\frac{(\varphi_0\circ\varphi'_a)^*\,\varphi'_a\circ\varphi_0}{\delta+\varepsilon_0-\varepsilon'_a+\frac{i\gamma}{2}},\quad \alpha=-\frac{6\pi\epsilon_0}{\omega^3}\frac{n\gamma/2}{\pi(\omega)^{-1}-n\Delta+in\Gamma/2},\quad S=-\frac{in(\gamma+\Gamma)/2}{\pi(\omega)^{-1}-n\Delta+in\Gamma/2}.\tag{29}$$

Since $\pi(\omega)$ is independent of $n$, the cooperative linewidth is linear in filling fraction $n$, consistent with the previous work [38].

Equation (24) and (29) together determine the cooperative optical response of the atomic array. In the next sections, we first validate our path-integral approach by showing that our result is consistent with previous literatures when the optical lattice is formed by a light with the magic wavelength. In this case, the trapping poential of the excited state is the same as that of the ground state. We then consider the effect of the trapping mismatch, which have been observed in the recent experimental realization of the atomic array [1].

## 3 Trapping Effects in Quantum Atomic Arrays

In this section, we analyze (24) and (29) in several limits. We first consider the case with $V_{e_{i_0}}(\mathbf{r})=V_g(\mathbf{r})$, and show that the result then matches the spin operator result. We also add comments on the relation between our approach and the Schwinger boson representation of spins [38, 39] [2]. We then study the effect of trapping mismatch, which leads imperfectness of mirrors and multiple resonances. We finally go beyond (24) and (29) by analyzing the recoil effect.

---

[2] We thank the Referee for bringing this work to our attention.

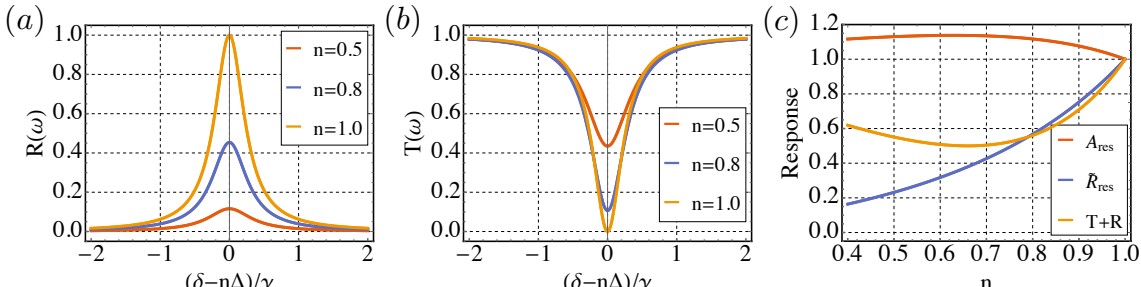

Figure 2: Numerical result for the fractional-filling effect with normal incident light with $\omega a_0 = 2\pi \times 0.68$. Here we take $V_{e_{i_0}}(\mathbf{r}) = V_g(\mathbf{r})$. (a). The reflection coefficient $R(\omega)$ as a function of detuning $\delta - n\Delta$ for different filling fraction $n$. (b). The transmission coefficient $T(\omega)$ as a function of detuning $\delta - n\Delta$ for different filling number $n$. (c). The filling-normalized absorptance $A$ and reflectance $\tilde{R}$, together with $T + R$ at cooperative resonance $\delta = n\Delta$, as a function of filling fraction $n$.

## 3.1 Magic Wavelength

We begin with the special case $V_{e_{i_0}}(\mathbf{r}) = V_g(\mathbf{r})$, which is valid when the optical lattice is formed by a light with the magic wavelength [46]. In this case, only the $a = 0$ term in (24) contributes and the transition of internal state does not couple to the motional degree of freedom. Moreover, there is no dependence on the detailed shape of the potential. This leads to

$$\pi(\omega) \equiv \frac{1}{\delta + \frac{i\gamma}{2}}, \qquad \alpha = -\frac{6\pi\epsilon_0}{\omega_0^3}\frac{n\gamma/2}{\delta - n\Delta + i(\gamma + n\Gamma)/2}, \qquad S = -\frac{in(\gamma + \Gamma)/2}{\delta - n\Delta + i(\gamma + n\Gamma)/2}. \tag{30}$$

Here we have assumed the normal incidence for the probe light. The cooperative linewidth becomes $\gamma + n\Gamma$, consistent with the experimental observation and numerical simulation in [1], and the previous work [38]. For $n < 1$, we find $|S| < 1$ even at the resonance and the mirror becomes imperfect. The transmission coefficient $T = |1+S|^2$ and reflection coefficient $R = |S|^2$ are found to be

$$T = \frac{(\delta - n\Delta)^2 + (1-n)^2\gamma^2/4}{(\delta - n\Delta)^2 + (\gamma + n\Gamma)^2/4}, \qquad R = \frac{n^2(\gamma + \Gamma)^2/4}{(\delta - n\Delta)^2 + (\gamma + n\Gamma)^2/4}. \tag{31}$$

The filling-normalized absorptance $A = (1-T)/n$ and reflectance $\tilde{R} = R/n$ can the be computed straightforwardly.

We plot the numerical result for $\omega a_0 = 2\pi \times 0.68$ as in the experiment [1] for various $n$ in Figure 2, where we have $\Delta/\gamma \approx 0.18$ and $\Gamma/\gamma \approx -0.48$. All above results reduces to the semi-classical results using spin operators when $n = 1$, where the frequency shift is $\Delta$ and the linewidth becomes $\gamma + \Gamma$. On the other hand, for $n \to 0$, we get back to the single-atom response with natural linewidth $\gamma$. As observed in the experiment [1], generally, we have $T + R < 1$. This is due to the fact that the self-energy of the excited state (18) contains the contribution of spontaneous emission of photons in arbitrary directions with random phases, which can not be observed by averaged $\mathbf{E}_{\text{tot}}$. However, the corresponding contribution exists if we measure energy density of electromagnetic field $\langle E^2(\mathbf{r})\rangle$ [34]. The filling-normalized absorptance $A$ show a weak dependence of $n$, while $\tilde{R}$ vanishes as $n \to 0$, consistent with the experimental observation and numerical simulation in [1].

Finally we comment on the relation between our results and the Swchinger boson/fermion representation of spins [38, 39]. Using (30), we find the bubble reads $\Pi_R(\omega) = d^2 n/(\delta + i\frac{\gamma(1-n)}{2})$. As we mentioned in the last section, the factor of $(1-n)$ exist due to the Pauli exclusion principle. This seems to be unphysical since after the ground state

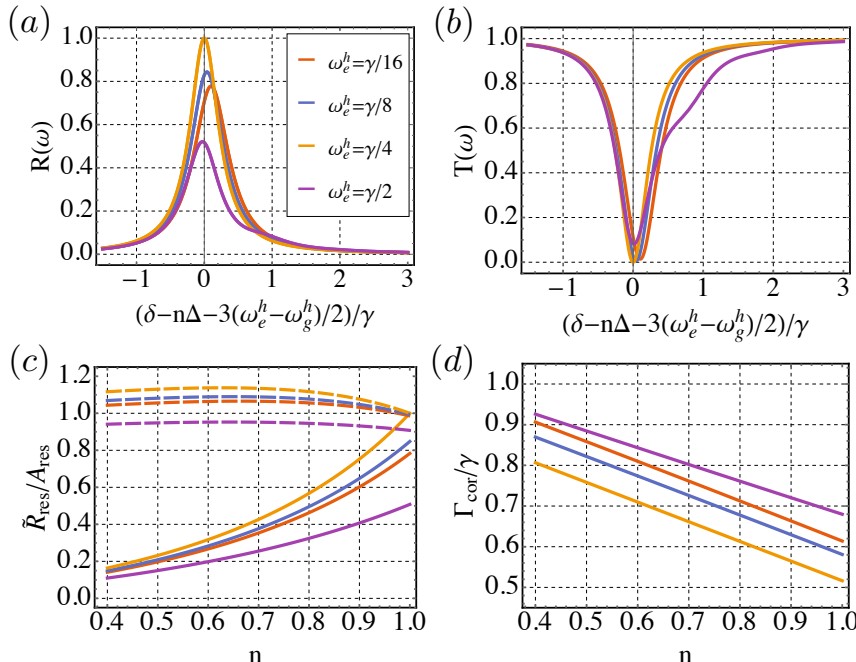

Figure 3: Numerical result for the trapping effect with normal incident light with $\omega a_0 = 2\pi \times 0.68$. We fix $\omega_g^h = \gamma/4$ and consider $\omega_e^h < \gamma$. (a). The reflection coefficient $R(\omega)$ as a function of detuning for $n = 1$ with different $\omega_e^h$. (b). The transmission coefficient $T(\omega)$ as a function of detuning for $n = 1$ with different $\omega_e^h$. (c). The fitted $A_{\mathrm{res}}$ and $\tilde{R}_{\mathrm{res}}$ as a function of $n$ for different $\omega_e^h/\omega_g^h$. Here the dashed lines corresponds to $A_{\mathrm{res}}$. (d). The fitted linewidth $\Gamma_{\mathrm{cor}}$ as a function of $n$ for different $\omega_e^h$.

particle being excited on some site, no Pauli exclusion factor is needed. However, the contribution from $-i\gamma n/2$ indeed cancels out with the corresponding contribution of the Green's function of photons $\tilde{\mathcal{G}}_{\mathrm{R}}^{\mathbf{E}}(\omega, \mathbf{k}_\parallel)$ in (29) due to (Recall that the definition of $\Gamma$ does not contain $\mathbf{r_n} = 0$, which is equal to $\gamma$.)

$$\mathbf{\Pi}_{\mathrm{R}}(\omega)^{-1} - \tilde{\mathcal{G}}_{\mathrm{R}}^{\mathbf{E}}(\omega, \mathbf{k}_\parallel) = \frac{1}{d^2 n}\left(\pi(\omega)^{-1} - \frac{i\gamma n}{2} + n\Delta + in\frac{\gamma+\Gamma}{2}\right) = \frac{1}{d^2 n}\left(\pi(\omega)^{-1} + n\Delta + in\frac{\Gamma}{2}\right). \quad (32)$$

This cancellation can be dated back to the cancellation between diagrams. Let's consider diagrams with one internal photons. Before contractions between $\psi_g$ and $\psi_g^\dagger$, it takes the form

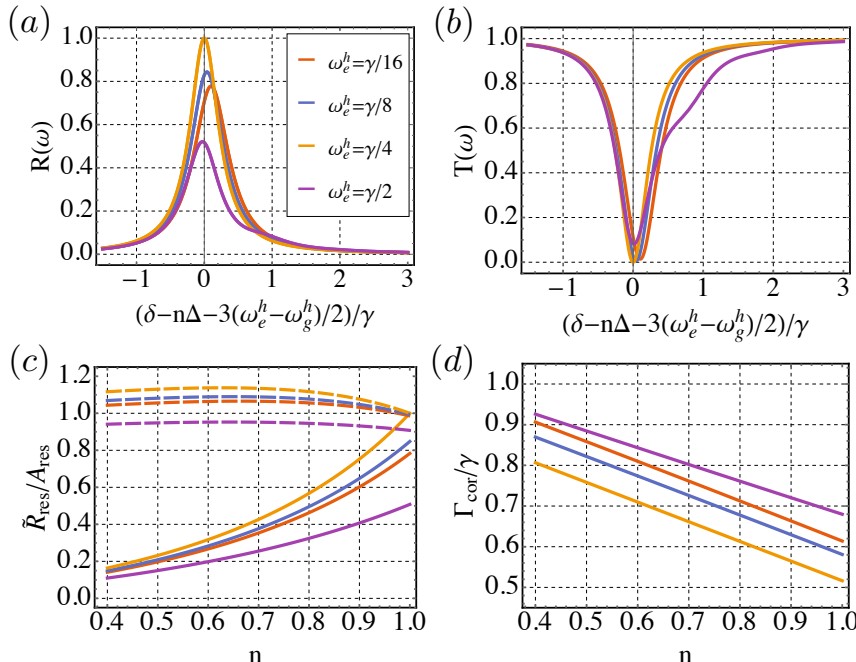

$$\begin{array}{cc}
\mathbf{r_n}, e_{i_0} & \mathbf{r_n}, e_{i_0} \\
\bullet\!\!\longleftarrow\!\!\bullet\!\!\sim\!\!\sim\!\!\bullet\!\!\longleftarrow\!\!\bullet & \\
\psi_g^\dagger(1) \quad \psi_g(1) \quad \psi_g^\dagger(2) \quad \psi_g(2) &
\end{array}. \quad (33)$$

If we contract $\psi_g(1)$ with $\psi_g^\dagger(2)$ and $\psi_g(2)$ with $\psi_g^\dagger(1)$, this leads to the diagram in self-energies which contains the unwilling factor of $(1-n)$. To realize the fact that when the ground state particle is excited by $\psi_g^\dagger(2)$, there is already no occupation due to $\psi_g(2)$, we also need to take into account the contribution by contracting $\psi_g(1)$ with $\psi_g^\dagger(1)$ and $\psi_g(2)$ with $\psi_g^\dagger(2)$. More explicitly, we have $n = \langle \psi^\dagger \psi \psi^\dagger \psi \rangle = \langle \psi^\dagger \psi \rangle \langle \psi^\dagger \psi \rangle + \langle \psi \psi^\dagger \rangle \langle \psi^\dagger \psi \rangle = n^2 + n(1-n) = n$. However, the new diagram is just the bubble diagram, which has been taken into account in our diagrammatic expansion (20). As a result, by summing up self-energies and bubbles, we find the correct result.

Due to this cancellation, we can alternatively drop the factor of $(1-n)$ in $\Pi_{\mathrm{R}}(\omega)$, and restrict the summation to $\mathbf{r} \neq 0$ in (26). This is then consistent with rules in [38, 39] using

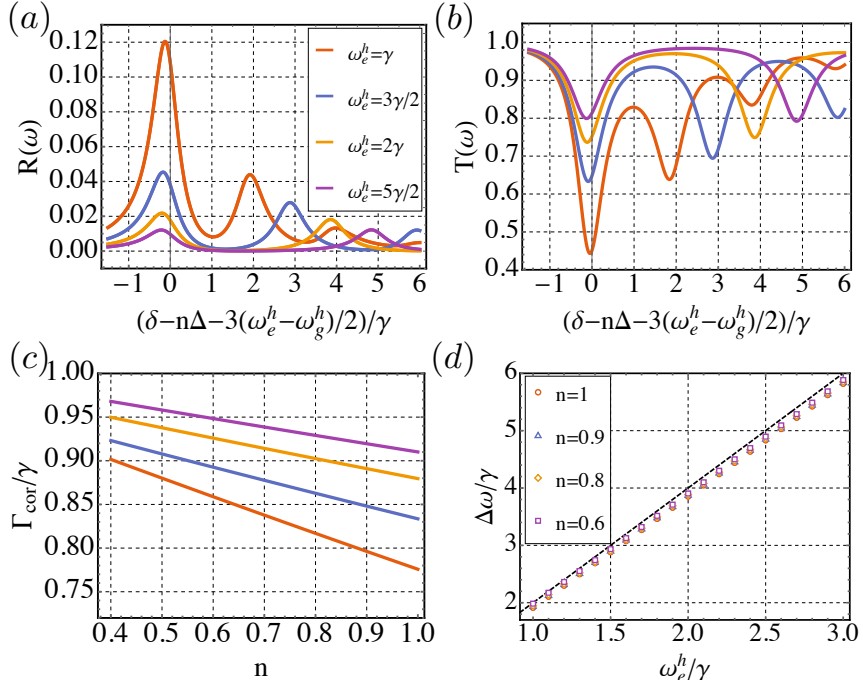

Figure 4: Numerical result for the trapping effect with normal incident light with $\omega a_0 = 2\pi \times 0.68$. We fix $\omega_g^h = \gamma/4$ and consider $\omega_e^h \geq \gamma$ in (a-c). (a). The reflection coefficient $R(\omega)$ as a function of detuning for $n = 1$ with different $\omega_e^h$. (b). The transmission coefficient $T(\omega)$ as a function of detuning for $n = 1$ with different $\omega_e^h$. (c). The fitted linewidth $\Gamma_{\text{cor}}$ as a function of $n$ for different $\omega_e^h$. (d). The fitted center of the second peak $\Delta\omega$ as a function of $\omega_e^h$ for different filling fraction $n$. Here the dashed line corresponds to $\Delta\omega = 2\omega_e^h$.

Schwinger particle representation, where atom on the same site can not appear twice. On the other hand, if the atoms are not trapped in optical lattices, the density-density correlation indeed plays an role [47]. In this case, the corresponding contribution in the self-energy should be kept. We give an example in Appendix A.

## 3.2 Trapping Mismatch

In this section, we discuss the effect of having $V_{e_{i_0}}(\mathbf{r}) \neq V_g(\mathbf{r})$. To make this problem analytically solvable, we expand the potential of near the minimum of each site and use the approximation of 3D isotropic harmonic potential. Ground-state atoms $|g\rangle$ and excited-state atoms $|e_{i_0}\rangle$ have a trapping frequency $\omega_g^h$ and $\omega_e^h$ correspondingly. The motional ground state wave function $\varphi_0(\mathbf{r})$ reads

$$\varphi_0(\mathbf{r}) = \left(\frac{\omega_g^h}{\pi}\right)^{\frac{3}{4}} e^{-\frac{\omega_g^h r^2}{2}}. \tag{34}$$

Under this approximation, the analytical expression for $\pi(\omega)$ can be obtained by relating $\pi(\omega)$ to the single-particle propagator in harmonic traps. The results are presented in Appendix B. It contains multiple resonances near $\delta = (3/2 + 2n)\omega_e^h - 3\omega_g^h/2$, broadened by the natural lifetime $\gamma$ of the excited state. For $\omega_e^h \gtrsim \gamma$, this leads to different peaks in the spectral $-\text{Im}\,\pi(\omega)/\pi$. For $\omega_e^h \lesssim \gamma$, different resonances merges, and only a single peak exists.

The parameters in the experiment [1] corresponds to $\omega_g^h < \gamma$, but at the same order $\sim$ MHz. We plot our results (29) for different $\omega_e^h/\gamma$, $\omega_g^h/\gamma$ and $n$ in Figure 3 and 4. We first fix

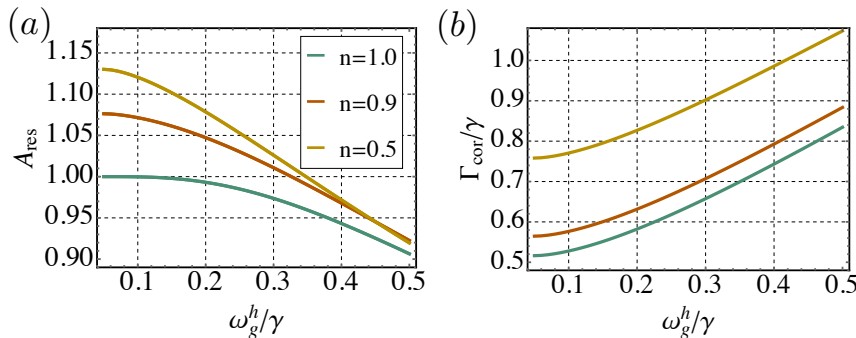

Figure 5: Numerical results for the effect of the trapping mismatch with normal incident light and $\omega a_0 = 2\pi \times 0.68$. We fix small excited state trapping frequency $\omega_e^h = \gamma/20$. (a). The fitted $A_{\text{res}}$ as a function of $\omega_g^h$ for different $n$. (b). The fitted linewidth $\Gamma_{\text{cor}}$ as a function of $\omega_g^h$ for different $n$.

$\omega_g^h/\gamma = 1/4$ and study the effect of $\omega_e^h \neq \omega_g^h$ for small $\omega_e^h < \gamma$. As shown in Figure 3 (a) and (b), both reflection coefficient $R(\omega)$ and transmission coefficient $T(\omega)$ show a single peak near $\delta - n\Delta - 3(\omega_e^h - \omega_g^h)/2 = 0$. For either $\omega_e^h > \omega_g^h$ or $\omega_g^h > \omega_e^h$, the atomic mirror becomes imperfect with max $R < 1$ and min $T > 0$. Motivated by the experimental result, we study the the cooperative linewidth of the atomic array by fitting the numerical result for $R(\omega)$ near $\delta = n\Delta + 3(\omega_e^h - \omega_g^h)/2$ as

$$R(\omega) = \frac{R_{\text{res}}\Gamma_{\text{cor}}^2/4}{(\delta - n\Delta - 3(\omega_e^h - \omega_g^h)/2 - \delta_0)^2 + \Gamma_{\text{cor}}^2/4}, \tag{35}$$

and define $T_{\text{res}} = T(n\Delta + 3(\omega_e^h - \omega_g^h)/2 + \delta_0)$. $\tilde{R}_{\text{res}}$ and $A_{\text{res}}$ can then be computed correspondingly using $R_{\text{res}}$ and $T_{\text{res}}$. The numerical results in (c-d) show $\tilde{R}_{\text{res}}$ and $A_{\text{res}}$ also decreases when $\omega_e^h \neq \omega_g^h$. The cooperative linewidth $\Gamma_{\text{cor}}$ is linear in $n$, with similar slope for different $\omega_e^h < \gamma$.

We then consider larger $\omega_e^h \geq \gamma$ in Figure 4. Now as shown in Figure 4 (a) and (b), multiply peaks appear in both reflection coefficient $R(\omega)$ and transmission coefficient $T(\omega)$. The center of peaks locates near energy $2n\omega_e^h$, where the transition from $|g\rangle$ to $|e_{i_0}\rangle$ is accompanied with the excitation of motional degree of freedom. As an example, we fit the position of the second peak $\Delta\omega$, and plot it as a function of $\omega_e^h$ in (d). Similar to the small $\omega_e^h$ case, the cooperative linewidth $\Gamma_{\text{cor}}$ is still linear in $n$. However, their slope show dependence on $\omega_e^h$.

We finally study $A_{\text{res}}$ and $\Gamma_{\text{cor}}$ as a function of $\omega_g^h$. We fix a small $\omega_e^h = \gamma/20$, as an analogy of the anti-trapped excited state in experiment [1], and tune $\omega_g^h$. As shown in Figure 5, we find when $\omega_g^h$ becomes larger, the absorption rate decreases and the cooperative linewidth becomes larger. This is due to the increase of the trapping mismatch for small $\omega_e^h$. For small $\omega_g^h$, the decrease in $A_{\text{res}}$ and the increase of the decay rate show quadratic dependence, while for large $\omega_g^h$, the dependence becomes linear. This is a close analogy of the experimental observation in [1].

## 3.3 Recoil Effects

Now we go beyond the limit of $\sigma \ll a_0$ and consider corrections to the leading order of $\eta = \sigma/a_0$ due to the recoil of atoms. The recoil effect has been discussed in previous works [40,41] using the Lindblad master equation.

We adopt the isotropic harmonic trap approximation used in the last subsection. In our approach, the recoil effects can be analyzed by using (4) without the approximation (8) as explained in the Appendix C. However, the same result can also be obtained using direct physical

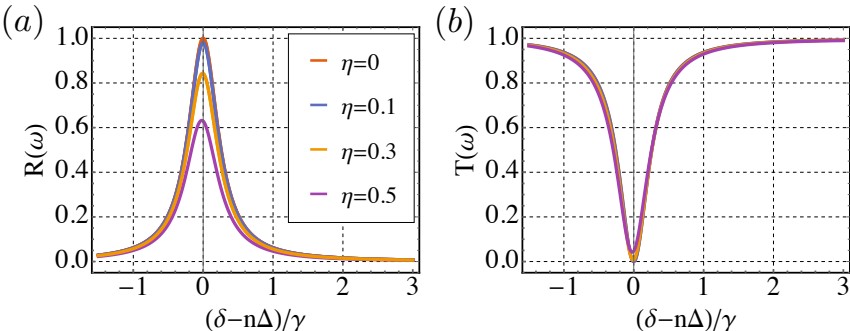

Figure 6: Numerical results for the recoil effect with normal incident light and $\omega a_0 = 2\pi \times 0.68$. We fix small excited state trapping frequency $\omega_g^h = 2\gamma$ and vary $\eta = \sigma/a_0$. (a). The reflection coefficient $R(\omega)$ as a function of detuning for $n = 1$ with different $\eta$. (b). The transmission coefficient $T(\omega)$ as a function of detuning for $n = 1$ with different $\eta$.

intuition. Our result (24) for $\pi(\omega)$ has a simple physical meaning: it measures the optical response of a single particle in the harmonic trap, with a lift time $\gamma$ for the excited state particle. Without the recoil effect, we take the inner product between wavefunctions $\varphi_0(\mathbf{r})$ and $\varphi_a'(\mathbf{r})$ to determine the transition rate for the motional degree of freedom. To take the recoil effect into account, we consider absorbing a photon with momentum $\mathbf{k}_1$ and emitting a photon with momentum $\mathbf{k}_2$. Physically, we expect the local response $\pi(\omega)$ takes the form

$$\pi_{\mathbf{k}_1 \mathbf{k}_2}(\omega) = \sum_a \int d\mathbf{r} d\mathbf{r}' \; \varphi_0(\mathbf{r})^* \varphi_a'(\mathbf{r}) e^{-i\mathbf{k}_2 \cdot \mathbf{r}} \frac{1}{\delta + \varepsilon_0 - \varepsilon_a' + \frac{i\gamma}{2}} e^{i\mathbf{k}_1 \cdot \mathbf{r}'} \varphi_a'(\mathbf{r}')^* \varphi_0(\mathbf{r}'). \quad (36)$$

Since the photons can propagate in any direction within $x - y$ plane, we further average over the direction of the momentum $\mathbf{k}_i = k(\cos\theta_i, \sin\theta_i, 0)$:

$$\pi(\omega) = \sum_a \int d\mathbf{r} d\mathbf{r}' \frac{d\theta_1}{2\pi} \frac{d\theta_2}{2\pi} \; \varphi_0(\mathbf{r})^* \varphi_a'(\mathbf{r}) e^{-i\mathbf{k}_2 \cdot \mathbf{r}} \frac{1}{\delta + \varepsilon_0 - \varepsilon_a' + \frac{i\gamma}{2}} e^{i\mathbf{k}_1 \cdot \mathbf{r}'} \varphi_a'(\mathbf{r}')^* \varphi_0(\mathbf{r}'). \quad (37)$$

We focus on the case with $\omega_g^h = \omega_e^h$. For small $\eta = \sigma/a_0 = k/\sqrt{\omega_g^h}$, we can expand $\pi(\omega)$ to obtain

$$\pi(\omega) = \frac{1}{\delta + i\gamma/2} - \frac{\eta^2/2}{\delta + i\gamma/2} + \frac{1}{8}\eta^4 \left( \frac{1}{\delta + i\gamma/2} + \frac{1/2}{\delta - 2\omega_g^h + i\gamma/2} \right) + O(\eta^6). \quad (38)$$

To the leading order $O(\eta^2)$, we find only a renormalization of the residue near $\delta + i\gamma/2 = 0$. At the sub-leading order, $O(\eta^4)$ we see a non-trivial contribution where atoms are excited due to the recoil even without trapping mismatch $\omega_e^h = \omega_g^h$, because the recoil increases the energy of atoms. However, when we further plot the reflection and transmission coefficients (see Figure 6), we find no visible peaks at $\delta - n\Delta = 2\omega_e^h$ even for $\omega_g^h > \gamma$ due to the suppression of $\eta^4$. We find $R(\omega)$ and $T(\omega)$ only show weak dependence on $\eta$ for small $\eta$. In particular, the parameter regime in the experiment [1] corresponds to $\eta \sim 0.1$, which is almost the same as $\eta = 0$. This justifies our discussions in previous sections.

## 4  Summary and Outlook

In this work, we study quantum atomic arrays using a microscopic model with atoms in optical lattices. We take a diagrammatic approach with PRA-like diagrams and obtain concise results for transmission and reflection coefficients. We find trapping mismatch can result in the imperfectness of mirrors. Multiple peaks exist when the local trapping frequency of the excited state $\omega_e^g \sim \gamma$. We also study the trapping frequency effects on the cooperative lifetime, and the effect of recoil for large but finite trapping frequency.

Our results can be tested in the experimental platforms similar to that in [1]. Recently, there are also experimental studies on the Pauli blocking of light scattering in degenerate fermions [48, 49]. The diagrammatic approach developed here can also be applied to study the optical response of degenerate fermion gases.

## Acknowledgments

We especially thank Yu Chen and Jianwen Jie for helpful discussions. We thank the Referee for bringing several related works to our attention, and the suggestion of studying the recoil effect. P.Z. acknowledges support from the Walter Burke Institute for Theoretical Physics at Caltech.

## A  Homogeneous Mirrors

Now we consider the case where the atom gas is homogeneous in $x-y$ plane at $z = 0$, similarly to [49]. The model of the full system reads

$$
\begin{aligned}
S = \int dt d\mathbf{r} \sum_i \psi_{e_i}^\dagger(t,\mathbf{r})\left(i\partial_t - \omega_0 + \frac{\nabla^2}{2}\right)\psi_{e_i}(t,\mathbf{r}) + \psi_g^\dagger(t,\mathbf{r})\left(i\partial_t + \frac{\nabla^2}{2}\right)\psi_g(t,\mathbf{r}) \\
+ \int dt d\mathbf{r} \sum_i |d|\left(\psi_{e_i}^\dagger(t,\mathbf{r})\psi_g(t,\mathbf{r}) + \psi_g^\dagger(t,\mathbf{r})\psi_{e_i}(t,\mathbf{r})\right)\mathbf{e}_i \cdot \mathbf{E}(t,\mathbf{r}).
\end{aligned}
\tag{39}
$$

The first step is again to determine the renormalization of the excited state Green's function. The self-energy reads

$$
\begin{aligned}
\Sigma_R^{e_i}(\tilde\omega, \mathbf{q}_\parallel)_{ij} = \overbrace{\phantom{xxxxx}}^{\mathbf{E} \quad e} = -\int \frac{d\mathbf{k}'}{(2\pi)^2}\frac{\omega^2 d^2}{\epsilon_0}\mathbf{e}_i \cdot \mathbf{G}(\omega, \mathbf{q}_\parallel - \mathbf{k}') \cdot \mathbf{e}_j (1 - n_F(\varepsilon_{k'})) \\
= -i\frac{\gamma}{2} + \int \frac{d\mathbf{k}'}{(2\pi)^2}\frac{\omega^2 d^2}{\epsilon_0}\mathbf{e}_i \cdot \mathbf{G}(\omega, \mathbf{q}_\parallel - \mathbf{k}') \cdot \mathbf{e}_j n_F(\varepsilon_{k'}) \equiv -i\frac{\gamma}{2}\delta_{ij} + \delta\mathcal{H}(\mathbf{q}_\parallel)_{ij}.
\end{aligned}
\tag{40}
$$

This leads to

$$
G_R^{e_i}(\tilde\omega, \mathbf{q}) = \frac{1}{\tilde\omega - \omega_0 - \varepsilon_q - \Sigma_R^{e_i}(\tilde\omega, \mathbf{q}_\parallel)} = \frac{1}{(\tilde\omega - \omega_0 - \varepsilon_q + i\frac{\gamma}{2})\mathbf{1} - \delta\mathcal{H}(\mathbf{q})}.
\tag{41}
$$

Here $\varepsilon_q = q^2/2$. We then compute the $\Pi_R$. The result reads

$$
\Pi_R(\omega, \mathbf{k}_\parallel) = \int \frac{d\mathbf{q}}{(2\pi)^2} \frac{d^2 n_F(\varepsilon_q)}{(\delta + \varepsilon_q - \varepsilon_{q+k} + i\gamma/2)\mathbf{1} - \delta\mathcal{H}(\mathbf{q} + \mathbf{k}_\parallel)}.
\tag{42}
$$

Summing up the contribution from photons, we find

$$\boldsymbol{\alpha}(\omega, \mathbf{k}_\parallel) = \frac{1}{-\boldsymbol{\Pi}_R(\omega, \mathbf{k}_\parallel)^{-1} - \omega^2 \mathbf{G}(\omega, \mathbf{k}_\parallel, z = 0)/\epsilon_0}. \tag{43}$$

Here since the system is homogeneous, the Fourier transform is

$$\mathbf{G}(\omega, \mathbf{k}_\parallel, z) = \int d\mathbf{r}_\parallel \, \mathbf{G}(\omega, \mathbf{r}) e^{-i\mathbf{k}_\parallel \cdot \mathbf{r}_\parallel} = \frac{i}{2k_z} e^{ik_z|z|} \mathcal{P}(\omega, \mathbf{k}_\parallel). \tag{44}$$

The relation between $\mathbf{E}_{\text{tot}}$ and $\alpha$ becomes

$$\begin{aligned}
\mathbf{E}_{\text{tot}} &= \mathbf{E}_0(\mathbf{r}) + \frac{\omega^2}{\epsilon_0} \int d\mathbf{r}' \mathbf{G}(\omega, \mathbf{r} - \mathbf{r}') \cdot \boldsymbol{\alpha}(\omega, \mathbf{k}_\parallel) \cdot \mathbf{E}_0 e^{i\mathbf{k}_\parallel \cdot \mathbf{r}'} \\
&= \mathbf{E}_0(\mathbf{r}) + \frac{\omega^2}{\epsilon_0} \mathbf{G}(\omega, \mathbf{k}_\parallel, z) \cdot \boldsymbol{\alpha}(\omega, \mathbf{k}_\parallel) \cdot \mathbf{E}_0 e^{i\mathbf{k}_\parallel \cdot \mathbf{r}}.
\end{aligned} \tag{45}$$

The $S$ matrix reads

$$\mathbf{S}(\omega, \mathbf{k}_\parallel) = \frac{i\omega^2}{2\epsilon_0 k_z} \mathcal{P}(\omega, \mathbf{k}_\parallel) \cdot \boldsymbol{\alpha}(\omega, \mathbf{k}_\parallel). \tag{46}$$

Again we consider the normal incident light. We further assume the density of the system is low as in [47]. We have

$$\begin{aligned}
\Pi_R(\omega, \mathbf{0})_{ij} &= \int \frac{d\mathbf{q}}{(2\pi)^2} \frac{d^2 n_F(\varepsilon_q)}{(\delta + i\gamma/2)\mathbf{1} - \delta\mathcal{H}(\mathbf{q})} \\
&= \frac{n_{2D} d^2}{\delta + i\gamma/2} \delta_{ij} + \frac{\omega^2 d^4}{\epsilon_0(\delta + i\gamma/2)^2} \int \frac{d\mathbf{q} d\mathbf{q}'}{(2\pi)^4} \left[ \mathbf{e}_i \cdot \mathbf{G}(\omega, \mathbf{q} - \mathbf{q}', z = 0) \cdot \mathbf{e}_j \right] n_F(\varepsilon_{q'}) n_F(\varepsilon_q) \\
&= \left( \frac{n_{2D} d^2}{\delta + i\gamma/2} + \delta\Pi_R(\omega) \right) \delta_{ij}.
\end{aligned} \tag{47}$$

We find

$$\alpha(\omega, \mathbf{0}) = \frac{1}{-\frac{\delta + i\gamma/2}{d^2 n_{2D}} + \frac{(\delta + i\gamma/2)^2}{d^4 n_{2D}^2} \delta\Pi_R(\omega) - \frac{i\omega}{2\epsilon_0}}, \quad S(\omega) = \frac{\frac{i\omega}{2\epsilon_0}}{-\frac{\delta + i\gamma/2}{d^2 n_{2D}} + \frac{(\delta + i\gamma/2)^2}{d^4 n_{2D}^2} \delta\Pi_R(\omega) - \frac{i\omega}{2\epsilon_0}}. \tag{48}$$

Here we have

$$\frac{(\delta + i\gamma/2)^2}{d^4 n_{2D}^2} \delta\Pi_R(\omega) = \frac{\omega^2}{\epsilon_0 n_{2D}^2} \int \frac{d\mathbf{q} d\mathbf{q}'}{(2\pi)^4} \left[ \mathbf{e}_i \cdot \mathbf{G}(\omega, \mathbf{q} - \mathbf{q}', z = 0) \cdot \mathbf{e}_i \right] n_F(\varepsilon_{q'}) n_F(\varepsilon_q). \tag{49}$$

This takes the similar form as results in [47] for the 3D case. This is the contribution from the density-density correlation in free fermion gases. Finally, we have

$$S(\omega) = \frac{\frac{i\omega}{2\epsilon_0} d^2 n_{2D}}{-\delta - i\gamma/2 + \frac{(\delta + i\gamma/2)^2}{d^2 n_{2D}} \delta\Pi_R(\omega) - \frac{i\omega}{2\epsilon_0} d^2 n_{2D}}, \tag{50}$$

which means $\delta\Pi_R(\omega)$ effectively shifts the resonant energy and the decay rate.

# B  The Analytical Formula for $\pi(\omega)$

In this Appendix, we present detailed derivation of the analytical formula for $\pi(\omega)$. We trick is to use the transformation to the time domain

$$
\begin{aligned}
\pi(\omega) &= \sum_a \int d\mathbf{r} d\mathbf{r}'\, \varphi_0(\mathbf{r})^* \varphi_a'(\mathbf{r}) \frac{1}{\delta + \varepsilon_0 - \varepsilon_a' + \frac{i\gamma}{2}} \varphi_a'(\mathbf{r}')^* \varphi_0(\mathbf{r}') \\
&= -\sum_a \int d\mathbf{r} d\mathbf{r}' \int_0^\infty d\tau\, \varphi_0(\mathbf{r})^* \varphi_a'(\mathbf{r}) e^{(\delta + \varepsilon_0 - \varepsilon_a' + \frac{i\gamma}{2})\tau} \varphi_a'(\mathbf{r}')^* \varphi_0(\mathbf{r}') \qquad (51) \\
&= -\int d\mathbf{r} d\mathbf{r}' \int_0^\infty d\tau\, e^{(\delta + \varepsilon_0 + \frac{i\gamma}{2})\tau} \varphi_0(\mathbf{r})^* K_{\omega_e^h}(\tau, \mathbf{r}, \mathbf{r}') \varphi_0(\mathbf{r}').
\end{aligned}
$$

Here we have assumed the integral over $\tau$ is convergent by restricting the $\delta + \varepsilon_0 < 3\omega_e^h/2$. After the integration, analytical continuation can be applied to release this restriction. Here $K_{\omega_e^h}(\tau, \mathbf{r}, \mathbf{r}')$ is the imaginary time Green's function in a harmonic trap with trapping frequency $\omega_e^h$. We have

$$
K_{\omega_e^h}(\tau, \mathbf{r}, \mathbf{r}') = \left( \frac{\omega_e^h}{2\pi \sinh \omega_e^h \tau} \right)^{\frac{3}{2}} \exp\left( -\frac{\omega_e^h}{2} \left[ (r^2 + r'^2) \coth \omega_e^h \tau - \frac{2\mathbf{r} \cdot \mathbf{r}'}{\sinh \omega_e^h \tau} \right] \right). \qquad (52)
$$

The integral over $\mathbf{r}$ and $\mathbf{r}'$ can be carried out first. We find

$$
\pi(\omega) = -2\sqrt{2} \int_0^\infty d\tau\, e^{a_0 \tau} \left( \frac{\omega_e^h \omega_g^h}{2\omega_e^h \omega_g^h \cosh \omega_e^h \tau + [(\omega_e^h)^2 + (\omega_g^h)^2] \sinh \omega_e^h \tau} \right)^{\frac{3}{2}}. \qquad (53)
$$

Here we have defined $a_0 = \left( \delta + \frac{3\omega_g^h}{2} + \frac{i\gamma}{2} \right)$ for conciseness. Then the integral over $\tau$ gives

$$
\pi(\omega) = \frac{\sqrt{2}}{\omega_e^h} \left( \frac{2\omega_e^h \omega_g^h}{(\omega_e^h + \omega_g^h)^2} \right)^{\frac{3}{2}} \frac{q^{-p-1} \left( (-2p(q-1) - q + 2) B_q\left(p + 1, \frac{1}{2}\right) - (2p + 3) B_q\left(p + 1, \frac{3}{2}\right) \right)}{1 - q}, \qquad (54)
$$

where $p \equiv -\frac{2a_0 + \omega_e^h}{4\omega_e^h}$ and $q \equiv \frac{(\omega_e^h + \omega_g^h)^2}{(\omega_e^h - \omega_g^h)^2}$. $B_z(a, b)$ is the incomplete beta function defined as $B_z(a, b) = \int_0^z t^{a-1}(1-t)^{b-1} dt$. For $\omega_e^h = \omega_g^h$, one can check that above result can be simplified as $\pi(\omega)^{-1} = a_0 - 3\omega_e^h/2$.

# C  The Derivation of the Recoil Effects

Here we give the derivation of (37) using the diagrammatic approach. For simplicity, we directly take diagrams under the rule of Schwinger bosons/fermions as discussed section 3.1. As a result, the self-energy of excited state $\Sigma_{\mathrm{R}}^{e_i}(q_0)$ is just a constant $-i\gamma/2$, and the summation in (20) is restricted by $\mathbf{r}_m \neq \mathbf{r}_n$.

To determine $\pi(\omega)$, we now examine a single diagram

$$\tag{55}$$

This corresponds to

$$i \int d\mathbf{r}_i \sum_m \mathbf{\Pi}_{\mathrm{R}}(\omega, \mathbf{r}', \mathbf{r}_3)\mathcal{G}_{\mathrm{R}}^{\mathrm{E}}(\omega, \mathbf{r}_{nm}+\mathbf{r}_3-\mathbf{r}_2)\mathbf{\Pi}_{\mathrm{R}}(\omega, \mathbf{r}_3, \mathbf{r}_2)\mathcal{G}_{\mathrm{R}}^{\mathrm{E}}(\omega, \mathbf{r}_{mp}+\mathbf{r}_2-\mathbf{r}_1)\mathbf{\Pi}_{\mathrm{R}}(\omega, \mathbf{r}_1, \mathbf{r}), \tag{56}$$

where we have

$$\mathbf{\Pi}_{\mathrm{R}}(\omega, \mathbf{r}, \mathbf{r}'')_{ii} = d^2 n\, \pi_i(\omega, \mathbf{r}, \mathbf{r}'') = \sum_a \varphi_0(\mathbf{r})^* \varphi'_{i,a}(\mathbf{r}) \frac{d^2 n}{\delta + \varepsilon_0 - \varepsilon'_{i,a} + \frac{i\gamma}{2}} \varphi'_{i,a}(\mathbf{r}')^* \varphi_0(\mathbf{r}'). \tag{57}$$

We have separated the integral over the full space into a summation over lattice sites $\mathbf{r}_n$, and an integral near each sites $\mathbf{r}_i$. For small $\sigma \ll a_0$, the dominate contribution comes from $\mathbf{r}_{mp} \gg \mathbf{r}_2 - \mathbf{r}_1$ and $\mathbf{r}_{nm} \gg \mathbf{r}_3 - \mathbf{r}_2$. Moreover, for normal (or nearly normal) incident light, we are probing the system with small $\mathbf{k}_{\parallel}$, which comes from contributions at large $\mathbf{r}_{nm}$ and $\mathbf{r}_{mp}$. We then expand $\tilde{\mathcal{G}}_{\mathrm{R}}^{\mathrm{E}}$ and take the standard approximation at long distance [50]:

$$\mathcal{G}_{\mathrm{R}}^{\mathrm{E}}(\omega, \mathbf{r}_{mp}+\mathbf{r}_2-\mathbf{r}_1) \approx \mathcal{G}_{\mathrm{R}}^{\mathrm{E}}(\omega, \mathbf{r}_{mp}) \exp(ik\hat{\mathbf{r}}_{mp} \cdot (\mathbf{r}_2-\mathbf{r}_1)). \tag{58}$$

Using this expression, we find

$$i \int d\mathbf{r}_3 d\mathbf{r}_1 \sum_m \mathbf{\Pi}_{\mathrm{R}}(\omega, \mathbf{r}', \mathbf{r}_3) e^{ik\hat{\mathbf{r}}_{nm}\cdot\mathbf{r}_3} \mathcal{G}_{\mathrm{R}}^{\mathrm{E}}(\omega, \mathbf{r}_{nm})\mathbf{\Pi}_{\mathrm{R}}^{nmp}(\omega)\mathcal{G}_{\mathrm{R}}^{\mathrm{E}}(\omega, \mathbf{r}_{mp}) e^{-ik\hat{\mathbf{r}}_{mp}\cdot\mathbf{r}_1} \mathbf{\Pi}_{\mathrm{R}}(\omega, \mathbf{r}_1, \mathbf{r}). \tag{59}$$

Here we have defined

$$\mathbf{\Pi}_{\mathrm{R}}^{nmp}(\omega)_{ii} = \sum_a \int d\mathbf{r}d\mathbf{r}'\, \varphi_0(\mathbf{r})^* \varphi'_{i,a}(\mathbf{r}) e^{-ik\hat{\mathbf{r}}_{nm}\cdot\mathbf{r}} \frac{d^2 n}{\delta + \varepsilon_0 - \varepsilon'_{i,a} + \frac{i\gamma}{2}} e^{ik\hat{\mathbf{r}}_{mp}\cdot\mathbf{r}'} \varphi'_{i,a}(\mathbf{r}')^* \varphi_0(\mathbf{r}'). \tag{60}$$

This already takes the form of (36), with the direction of the photons determined by $\hat{\mathbf{r}}_{nm}$ and $\hat{\mathbf{r}}_{mp}$. Note that due to the rotation symmetry of the isotropic harmonic trap, it only depends on $\hat{\mathbf{r}}_{nm} \cdot \hat{\mathbf{r}}_{mp}$. Finally, we need to perform the Fourier transform by summing up $m, n, p$. For large $\mathbf{r}_{nm}$ and $\mathbf{r}_{mp}$, we approximate this summation as a average over $\hat{\mathbf{r}}_{nm} \cdot \hat{\mathbf{r}}_{mp}$. Finally, we obtain the "naive" formula (37) by focusing on incident light polarized in the direction of $i_0$.

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
