# Peer review of "Trapping Effects in Quantum Atomic Arrays"

_SciPost Physics, doi:SciPost Phys. Core 5, 010 (2022)_

## Round 3 · Referee Report · Anonymous (Referee 1) · 2022-2-7

Report

I am satisfied with the revision of the manuscript and with how my comments have been addressed. In particular, I appreciate the effort of the Authors to connect to previous results and justify their diagrammatic approach. The expanded analysis of trapping effects is also important, as it provides a number of qualitative predictions to be verified in experiment.

While the arguments are still understandable, the number of language mistakes is very high.

---

## Round 3 · Referee Report · Anonymous (Referee 2) · 2022-2-17

Report

After the first round of revisions, the author has improved the style and presentation of the draft. These changes include a change of the title, the corrections of many typos and mistakes especially in Section 2.1, as well as adding a section about "Recoil Effects". In addition the author now provides two new appendices: Appendix A studies the scattered light from a homogeneous Fermi gas in the x-y plane and Appendix C provides more details about the recoil effect. In the new section "Recoil Effects" the author shows and estimates the transition probability between different bands even in absence of a trapping mismatch. It is shown that recoil effects are usually negligible and can be treated and added perturbatively.

As I have stated in my previous report, I believe that the topic of this work is timely and the results are useful for the specialized physics community working with atomic arrays. The new section extends and supports nicely the previously presented results. However, in their reply as well as in their manuscript the author was not able to give me a reason to believe that this work is (1) a groundbreaking theoretical discovery, (2) a breakthrough on a long-standing research stumbling block, (3) opening a new pathway in a research direction, or (4) providing a synergetic link of different research areas. Thus, I still believe, this manuscript does not meet the "expectations" for a paper to be accepted in SciPost Physics. I totally agree with the author that the trapping effect is relevant for practical reasons. Nevertheless, the author does not study "new physics" but rather how trapping effects modify previously described physics. Since the study of fractional filling and trapping mismatch is interesting for the specialized community the author should consider to submit to a more specialized journal.

About the draft: despite the improvements, I still find many mistakes and typos in the manuscript, basically on every page including the Appendices. Also, while I could reproduce some of the results especially the ones discussed in Sec. 3.1 and Sec. 3.2, I still find it extremly hard to follow the line of thought in Sec. 2.2. While there might be no mistakes in the calculations, there are still many Greensfunctions and operations undefined, making it basically impossible to reproduce the results without reading through the existing and cited literature. This lack of "clarity" is another even more general problem of this draft.

---

## Round 3 · Author Response

Dear Editor and Referees,

We would like to thank you for handling and reviewing our manuscript. We have revised our manuscript, taking care of all comments of both referees. Here we resubmit the paper with a new title, ``Trapping Effects in Quantum Atomic Arrays'', for your consideration.

We are glad to see both referees find our study of trapping mismatch interesting and provide suggestions on improving our manuscript. In this revised version, we considerably extend and rewrite most parts of the manuscript, adding new calculations, new discussions, and new figures. Referee 1 mainly concerns about the diagrams that are taken into account in this work. We thank Referee 1 for bringing previous works to our attention. As explained below and in the revised manuscript, we show the consistency between our approach and previous works by direct calculation. We further explain it as a cancellation between diagrams. Referee 1 also suggests adding calculations on the recoil effects. This is now analyzed in section 3.3.

Referee 2 pointed out several typos in the manuscript. We thank Referee 2 for carefully examining our manuscript, and we are sorry for the typos in the previous version. In this revised version, we have improved the presentation of the results by correcting typos, adding explanations, and changing some names of variables throughout the whole manuscript. We hope Referee 2 finds this revised version satisfactory. We also hope to point out that our work studies the interesting trapping effect. This is directly motivated by experiments and is inevitable for any experimental realization of the quantum atomic array, which has the potentials for realizing novel quantum phases/dynamics and building quantum memories. As a result, we believe our work is appropriate for getting published in SciPost.

In the following, we give a detailed response to the referees' reports. We believe that after the extensive improvements, our manuscript is now suitable for publication. We hope you find our revision satisfactory. We appreciate your efforts in reconsidering our manuscript.

Sincerely yours,

Pengfei Zhang

Reply to Referee 1

We are glad to see Referee 1 find our work interesting and would like to suggest its publication. Now we have extensively revised our paper by rewrite many parts of the paper based on the comments of Referee 1. We added discussions on the diagrammatic method, extended the analysis of trapping mismatch including adding calculations on the recoil effects, and corrected typos. Here are some detailed response:

  1. We thank Referee 1 for bringing our attention to all these previous papers that are closely related to our work. In this revised version, we add several discussions between our work and previous results. Since the linear dependence of filling fraction has been pointed out in previous works, we now change the title of the paper, focusing on the trapping effects in the main text.

  2. We thank Referee 1 for his/her expert's comments on the diagrammatic approach with new Feynman rules. We indeed find our result is consistent with previous works. Firstly, we add a calculation, directly showing that we can keep the $(1-n)$ factor in the self-energy (such a factor is due to the fact that in many-body diagrammatic expansion, the Green's function corresponds to adding an additional particle on the top of the many-body state) and obtain the correct result as in Eq. (32). An important reason is that, unlike previous works, in our approach, the summation defined in Eq. (26) contains the contribution from $\mathbf{r}_n=0$. Such a contribution cancels with the $(1-n)$ factor. This can also be understood as a cancellation between diagrams due to indistinguishable particles, as explained in Eq. (33) and discussions below. We also give an example that the factor of $(1-n)$ can lead to physical results in a different setup. Such an example is given in the new Appendix A.

  3. We thank Referee 1 for the suggestion of analyzing the recoil effects. In this revised version, we consider the leading order effects due to the recoil of atoms in section 3.3. The result indeed shows for $\sigma/a_0\sim 0.1$ (as in experiment), the correction is small. As a result, this analysis validates our discussions in previous sections. We also add the suggestion of the Referee in our acknowledgments.

  4. We have corrected typos and errors throughout the manuscript.

We hope Referee 1 find these satisfactory and would like to suggest the publication of our work in SciPost.

Reply to Referee 2

We thank Referee 2 for carefully reading our manuscript and examining all equations. We are sorry for the typos in the previous version. In this revised version, we have improved the presentation of the results by correcting typos, adding explanations, and changing some names of variables throughout the whole manuscript. We hope Referee 2 finds our revised manuscript satisfactory.

We respectively disagree that the conclusion of Referee 2 that our work, although ``timely and the results are useful for the physics community working with atomic arrays'', is not enough for getting published in SciPost. Controlling the interaction between atoms and photons is one of the most important topics in modern quantum physics, and atomic arrays are a plausible new experimental related platform for the study. It has the potentials for realizing novel quantum phases/dynamics and building quantum devices, including quantum memories. Our work directly analyzed the trapping effects, which is inevitable for experiments, and thus an important contribution to the field. As a result, we believe our work should be published in SciPost.

---

## Round 3 · List of Changes

1. We considerably extend and rewrite most parts of the manuscript, adding new calculations, new discussions, and new figures.
  2. We improved the presentation of the results by correcting typos, adding explanations, and changing some names of variables throughout the whole manuscript.

---

## Editorial Decision

published